# Development and Evaluation of a Robust Sandwich Immunoassay System Detecting Serum WFA-Reactive IgA1 for Diagnosis of IgA Nephropathy

**DOI:** 10.3390/ijms23095165

**Published:** 2022-05-05

**Authors:** Yuta Uenoyama, Atsushi Matsuda, Kazune Ohashi, Koji Ueda, Misaki Yokoyama, Takuya Kyoutou, Kouji Kishi, Youichi Takahama, Masaaki Nagai, Takaaki Ohbayashi, Osamu Hotta, Hideki Matsuzaki

**Affiliations:** 1Reagent Engineering, Protein Technology Group, Sysmex Corporation, Kobe 651-2271, Hyogo, Japan; Uenoyama.Yuta@sysmex.co.jp (Y.U.); Matsuda.Atsushi@sysmex.co.jp (A.M.); Ohashi.Kazune@sysmex.co.jp (K.O.); Ueda.Koji@sysmex.co.jp (K.U.); Yokoyama.Misaki@sysmex.co.jp (M.Y.); Kyoutou.Takuya@sysmex.co.jp (T.K.); Takahama.Youichi@sysmex.co.jp (Y.T.); 2Bio-Reagent Material Development, Bio-Diagnostic Reagent Technology Center, Sysmex Corporation, Kobe 651-2271, Hyogo, Japan; Kishi.Kouji@sysmex.co.jp; 3Division of Nephrology, Narita Memorial Hospital, Toyohashi 441-8029, Aichi, Japan; nagai5823@meiyokai.or.jp (M.N.); 69990000@meiyokai.or.jp (T.O.); 4Division of Internal Medicine, Hotta Osamu Clinic (HOC), Sendai 984-0013, Miyagi, Japan; hottao@remus.dti.ne.jp

**Keywords:** IgA nephropathy, immunocomplex, glyco-diagnosis, lectin, lectin microarray, automated immunoassay, *Wisteria floribunda* agglutinin, agglutinated IgA1

## Abstract

Aberrant glycosylation of IgA1 is involved in the development of IgA nephropathy (IgAN). There are many reports of IgAN markers focusing on the glycoform of IgA1. None have been clinically applied as a routine test. In this study, we established an automated sandwich immunoassay system for detecting aberrant glycosylated IgA1, using *Wisteria floribunda* agglutinin (WFA) and anti-IgA1 monoclonal antibody. The diagnostic performance as an IgAN marker was evaluated. The usefulness of WFA for immunoassays was investigated by lectin microarray. A reliable standard for quantitative immunoassay measurements was designed by modifying a purified IgA1 substrate. A validation study using multiple serum specimens was performed using the established WFA-antibody sandwich automated immunoassay. Lectin microarray results showed that WFA specifically recognized *N*-glycans of agglutinated IgA1 in IgAN patients. The constructed IgA1 standard exhibited a wide dynamic range and high reactivity. In the validation study, serum WFA-reactive IgA1 (WFA+-IgA1) differed significantly between healthy control subjects and IgAN patients. The findings indicate that WFA is a suitable lectin that specifically targets abnormal agglutinated IgA1 in serum. We also describe an automated immunoassay system for detecting WFA+-IgA1, focusing on N-glycans.

## 1. Introduction

IgA nephropathy (IgAN) is the most common primary glomerulonephritis worldwide and is one of the causes of end-stage renal disease [1]. IgAN is an immune-mediated disease that involves deposition of abnormally agglutinated IgA1 that has polymerized or formed an immune complex in the glomerular mesangium, followed by the onset of mesangioproliferative glomerulonephritis [2,3], an essential pathological feature of IgAN is IgG and/or IgA autoantibodies to aberrant glycosylated IgA1 [4]. Although the treatment of IgAN has not yet been clearly defined [5], tonsillectomy and steroid pulse therapy at a relatively early stage are beneficial for clinical remission in IgAN patients [6]. Therefore, comparative diagnosis at an early stage is important to improve the prognosis of patients with IgAN. Kidney biopsy is routinely performed for a definitive diagnosis [7]. Serum IgA and complement 3 (C3) levels, or the ratio of IgA1/C3, are often used routinely as a serological diagnostic test for IgAN. Unfortunately, these indicators are not satisfactory for precise IgAN diagnosis in terms of sensitivity and specificity [8]. Therefore, to improve prognosis, a new highly accurate and noninvasive diagnostic procedure is needed.

Aberrant glycosylation of IgA1 is involved in the pathologic mechanism of IgAN [9]. IgA1 consists of a hinge region that contains several *O*-glycosylation sites between the first and second constant-region domains of the heavy chain. Alteration of the glycoforms in the hinge region is essential for IgAN because the immune multicomplex is caused by specific autoantibodies binding to the aberrant *O*-glycosylation in the hinge region of IgA1 [2,3,4]. The development of diagnostic procedures for IgAN that focus on aberrant *O*-glycosylation has been promoted. Notably, lectin enzyme-linked immune sorbent assay (ELISA) and detection of autoantibodies against the abnormal *O*-glycosylation of IgA1 have been investigated to differentiate IgAN from non-IgAN patients and healthy donors [10,11,12,13]. Recently, antibodies with specificity to serine/threonine-linked N-acetylgalactosamine (GalNAc) residues in the hinge region of IgA1 (galactose-deficient IgA1, Gd-IgA1) were developed as suitable serological markers for IgAN [14,15]. Studies using various cohorts have shown that this antibody is superior in discriminating IgAN from healthy donors [16,17,18]. However, even with this antibody, it is still difficult to accurately and sensitively diagnose IgAN without kidney biopsy.

While the O-glycosylation of IgA1 has been described, there are few reports on *N*-glycosylation [19,20,21]. Among them, Narimatsu et al. reported that *Wisteria floribunda* agglutinin (WFA) binds to terminal GalNAc [22], and specifically recognizes an abnormal agglutinated serum IgA1 in monoclonal immunoglobulin deposition disease associated with monoclonal IgA [23]. The authors also implicated serum WFA-reactive IgA1 (WFA+-IgA1) as a candidate for monitoring IgA1 agglutination in the bloodstream. The same study also showed that WFA recognizes *N*-glycans, rather than *O*-glycans, on IgA1, based on the disappearance of the interaction between WFA and agglutinated IgA1 after peptide: N-glycosidase (PNGase) treatment of IgA1. The findings suggest that the mechanism of WFA binding to IgA1 is enhanced with multivalency due to multivalent IgA1 harboring the asialo-*N*-acetyl-lactosamine (LacNAc) structure, because previous studies have not reported terminal GalNAc on the *N*-glycans of serum IgA1. However, it has not been verified that WFA selectively binds to serum-agglutinated IgA1 in IgAN patients. Furthermore, it is unclear whether serum WFA+-IgA1 is suitable as a diagnostic marker of IgAN, because validation studies using multiple specimens have not been performed. Therefore, it is essential to establish an automated immunoassay system capable of detecting serum WFA+-IgA1 and evaluating WFA+-IgA1 as an IgAN diagnostic marker through a validation study using multiple serum specimens.

Previously, we reported the development of an automated chemiluminescent enzyme immunoassay (CLEIA) system to detect serum WFA-reactive Mac 2 binding protein (WFA+-M2BP) [24]. WFA+-M2BP is currently used clinically as a suitable indicator of liver fibrosis in hepatic C virus infection [25,26,27]. Based on this, we attempted to measure serum WFA+-IgA1 using this detection system and to evaluate the clinical performance. The results demonstrate that WFA specifically binds to agglutinated IgA1, rather than monomers, in IgAN patients. The optimal standard for the WFA+-IgA1 sandwich automated CLEIA system was designed and implemented. Finally, the automated immunoassay was validated.

The reliable, reproducible, and automated CLEIA system is capable of high throughput with a short reaction time. WFA, which specifically recognizes abnormally agglutinated IgA1 in IgAN patients, was validated as an IgAN marker of WFA+-IgA1 in the automated WFA-antibody sandwich CLEIA system using serum from IgAN, non-IgAN, and healthy control (HC) subjects.

## 2. Results

### 2.1. WFA Binding to Abnormal Agglutinated IgA1

In a previous study, the importance of WFA+-IgA1 as a diagnostic marker for IgAN was suggested by observing that WFA specifically recognizes agglutinated IgA1, rather than a monomer [22]. We verified this immunoprecipitation of IgA1 with anti-IgA1 monoclonal antibody (mAb) and WFA from the serum of three HCs and patients with IgAN (Figure 1). Purified IgA1 from each specimen was isolated using non-reducing sodium dodecyl sulfate polyacrylamide gel electrophoresis (SDS-PAGE) and visualized by silver staining and Western blotting with anti-IgA1 antibody. IgA1 that was immunoprecipitated with anti-IgA1 mAb was not significantly different between the specimens. Most serum IgA1 was resolved as a monomer of approximately 150kDa (Figure 1a). The expression pattern of IgA1 after immunoprecipitation with WFA was different from that of IgA1-immunoprecipitation (Figure 1b). Interestingly, these bands were confirmed in highly molecular regions, particularly in some patients with IgAN. These findings indicate that multimerized IgA1 was specifically captured by WFA and that WFA recognizes an immunocomplex that includes IgA1 of IgAN patients compared to that of HCs.

### 2.2. Verification of WFA Detection of Abnormally Agglutinated IgA1 Using the Automated Chemiluminescent Assay

The results in Figure 1 suggest that WFA recognizes the immunocomplex, including IgA1 derived from IgAN patients. Next, glycan analysis of IgA1 was performed to evaluate the suitability of WFA for the construction of an automated CLEIA system. An antibody-overlay lectin microarray was performed for differential glycan analysis of IgA1 derived from three HCs and three patients with IgAN (Figure 2). Purification of IgA1 from each specimen was confirmed by SDS-PAGE and silver staining under reducing conditions (Figure 2a). Few contaminants could compete on the lectin microarray, and there was no difference in IgA1 quantification. Subsequently, the immunoprecipitated samples were analyzed using the lectin microarray. Figure 2b shows the typical images of microarray results in HC and IgAN patients (the layout of the arrayed lectins is shown in Appendix A). The WFA signal increased significantly in IgA1 derived from IgAN patients compared with that in HCs (Figure 2c). Moreover, the WFA signal in IgAN patients was completely disrupted by N-glycan removal using PNGase (Appendix A). Next, the correlation of WFA signals with lectin microarray and WFA+-IgA1 CLEIA counts was investigated (Figure 2d). Each value was significantly correlated between these measurements. The findings suggest that the level of abnormally agglutinated IgA1 with N-glycans that bind to WFA increased specifically in IgAN patients and that WFA+-IgA1 could be detected with the automated CLEIA system.

### 2.3. Construction of Reliable Standard IgA1 for the Automated CLEIA

To evaluate serum WFA+-IgA1 as an IgAN diagnostic marker, it is important to construct a suitable detection system using an automated analyzer for validation studies. To achieve this, a reliable standard is required for quantitative analysis. Since WFA specifically recognizes polymeric IgA1, we designed two types of IgA1 standards and evaluated their reactivity in CLEIA (Figure 3a). One is monomeric IgA1 (nIgA1), and the other is polymeric IgA1 (Glt-IgA1), in which molecules are cross linked with glutaraldehyde. The constructed IgA1 standards were evaluated using SDS-PAGE and size exclusion chromatography (SEC) (Figure 3b). Bands and peaks of Glt-IgA1 were confirmed in the high-molecular region compared with nIgA1 in SDS-PAGE and SEC analysis, respectively. The binding activity of the constructed Glt-IgA1 to WFA was investigated using lectin microarray analysis. Since removal of sialic acid by sialidase digestion increased the WFA signal (Appendix A), lectin microarray analysis was performed on nIgA1 without sialidase digestion and Glt-IgA1 with or without sialidase digestion (Figure 3c and Appendix A). The binding of WFA to the constructed Glt-IgA1 was higher than that of nIgA1. The WFA binding signal was further enhanced by sialidase-mediated sialic acid removal. These results indicate the successful construction of a standard suitable for automated CLEIA analysis of serum WFA+-IgA1. The results further suggest that WFA binds more strongly to agglutinated IgA1. Subsequently, we evaluated the availability of the constructed standard IgA1 for automated CLEIA analysis. The evaluation was performed using the chemiluminescent count obtained using an automatic CLEIA analyzer (Figure 3d). The chemiluminescent count for Glt-IgA1 increased significantly compared to that for nIgA1. These counts were further enhanced by sialidase treatment. The result correlated well with the WFA signal pattern of the lectin microarray analysis. Finally, the CLEIA count of Glt-IgA1 with sialidase treatment was approximately 700-times higher than that of nIgA1 not treated with sialidase. In addition, the linear response of Glt-IgA1 with the CLEIA system had a wide dynamic range (0–100,000 ng/mL as Glt-IgA1, R^2^ = 0.9998; Figure 3e). With these results, we successfully constructed the available IgA1 standard for an automated CLEIA analyzer.

### 2.4. Evaluation of Serum WFA+-IgA1 as IgAN Diagnostic Marker with Automated CLEIA

An available automated immunoassay for the detection of serum WFA+-IgA1 is shown in Figure 3. Subsequently, the clinical diagnostic performance of serum WFA+-IgA1 for IgAN was evaluated using the automated CLEIA system. Serum WFA+-IgA1 levels were measured in 50 HCs, 43 non-IgAN patients, and 47 IgAN patients before tonsillectomy (Figure 4a). In addition, we compared serum WFA+-IgA1 levels before and after tonsillectomy in 38 IgAN patients (median days after tonsillectomy, 184; range, 35–743) (Figure 4b). Serum WFA+-IgA1 levels were significantly different between HCs and IgAN patients (*p* < 0.05). However, there was no significant difference between non-IgAN and IgAN patients. Serum WFA+-IgA1 levels decreased significantly after tonsillectomy compared to those before tonsillectomy (*p* < 0.01). Moreover, many IgAN patients showed a marked decrease in WFA+-IgA1 counts during follow-up after tonsillectomy (Appendix A). These results suggest that WFA+-IgA1 counts respond to renal lesions and are useful for monitoring the therapeutic effects of IgAN.

### 2.5. Comparison of IgAN Diagnostic Serological Markers

As shown in Figure 4, we successfully constructed an automated immunoassay system for serum WFA+-IgA1 detection and evaluated its clinical performance using multiple clinical samples. Next, we investigated the diagnostic performance and characteristics of WFA+-IgA1 by comparing WFA+-IgA1 with other IgAN serological markers, such as Gd-IgA1, total IgA, and C3 (Table 1 and Appendix A). In the comparison between 50 HCs and 47 IgAN patients, all markers showed statistically significant differences. In contrast, in the comparison between non-IgAN and IgAN patients, although total IgA, IgA/C3, and Gd-IgA1 showed significant differences, WFA+-IgA1 and C3 did not. Among them, the Gd-IgA1 marker displayed the most significant difference between HCs and IgAN, non-IgAN, and IgAN patients (*p* < 0.01). The results implicate Gd-IgA1 as the most suitable marker for IgAN diagnosis.

### 2.6. Comparative Analysis of Combined Markers with WFA+-IgA1 and Other IgAN Markers

A correlation analysis of WFA+-IgA1 was performed to investigate the properties of WFA+-IgA1. Interestingly, the levels of Gd-IgA1, total IgA, and C3 deviated from the WFA+-IgA1 value (Appendix A). In particular, dissociation appeared to be prominent in patients with non-IgAN and IgAN. The findings suggest that WFA+-IgA1 has properties that differ from those of the other markers in the diagnosis of IgAN. Based on these results, we explored the possibility of improving IgAN diagnostic performance using a combination of WFA+-IgA1 and other markers. The results of the receiver operating characteristic (ROC) analysis of each single marker and each combination were compared as the area under the ROC curve (AUC) value (Table 2 and Appendix A). ROC analysis showed that Gd-IgA1 had the best diagnostic value as a single marker (AUC = 0.734). In the combination analysis, the AUC values of the combination of WFA+-IgA1 and Gd-IgA1 (WFA-Gd), WFA+-IgA1 and total IgA (WFA-Total), WFA+-IgA1, and Gd-IgA1 and C3 (WFA-Gd/C3) were 0.748, 0.660, 0.704, and 0.789, respectively. In particular, the combination of WFA-Gd and WFA-Gd/C3 exceeded the IgAN diagnostic performance of Gd-IgA1. The collective findings suggest that the combined test of indices with different characteristics is suitable for IgAN diagnosis and that WFA+-IgA1 is a useful marker that can complement Gd-IgA1.

## 3. Discussion

The aberrant glycosylation of proteins is associated with the pathological features of various diseases. The glycosylation of glycoproteins is an attractive target for the development of diagnostic and therapeutic agents. With recent technological innovations, such as mass spectrometry and microarray, many comprehensive studies for biomarker development associated with alternative protein glycosylation are being performed using a modern glycoproteomics approach [28]. In IgAN, IgA1 deposition in the glomerulus is triggered by increased serum IgA1 with aberrant O-glycosylation in the hinge region. Many attempts to develop diagnostic markers targeting the glycosylated region have failed to identify markers that have reached clinical application. Sandwich immunoassays using lectins and antibodies against core proteins are often used to detect glycoproteins with disease-specific glycosylation as diagnostic targets. A sandwich ELISA with *Helix aspersa* agglutinin, Tn antigen (GalNAc) binding lectin, and anti-IgA1 antibody has been constructed to detect altered glycosylation of IgA1 in IgAN patients, and many clinical evaluations have been reported [11,12,14]. In other diseases, attempts to detect the α-fetoprotein-L3 hepatocellular carcinoma marker by sandwich ELISA using fucose-binding lectins have been reported [29]. However, the sensitivity and specificity of lectin-antibody sandwich detection is unsatisfactory for disease diagnosis. In general, the interaction of these lectins with their glycan ligands is typically lower than that of antibody–protein interactions, with *K*_d_ values in the M^−7^ to M^−3^ range [30,31]. In addition, the binding specificity of lectins is broader than that of the antibodies. These properties make it difficult to construct a simple and readily available detection system for a few glycoproteins with aberrant glycosylation in the bloodstream.

Several antibodies to glycopeptide containing glycan structures have been recently described [15,32,33,34]. These studies indicate that glycoproteins modified with glycans can be detected without the use of lectins. The development of such antibodies could be valuable for the highly sensitive and reproducible detection of disease-specific aberrant glycosylated glycoproteins. However, most of these unique antibodies recognize *O*-glycosylated glycopeptides. In contrast, although there are some reports of antibodies that recognize glycopeptides with *N*-glycans [35], the number of reports is considerably less compared with that of glycopeptides with *O*-glycans. This is because the *N*-glycan structure is more complex and diverse than that of *O*-glycan. Therefore, sandwich immunoassays using lectins are generally used to detect *N*-glycosylated glycoproteins.

As mentioned above, it seems more difficult to construct an immunoassay with lectins, rather than antibodies. Nonetheless, there have been some reports of successful construction of highly sensitive immunoassays using lectins [36,37,38]. Interestingly, these reports demonstrated that lectins are immobilized and that their multivalency provides high reactivity to glycan epitopes. As a property of lectins, firmer interactions are often formed by the simultaneous interaction of multiple lectins with the target ligands [30,31,39]. In this study, despite targeting *N*-glycans on IgA1, we established an automated CLEIA system that can rapidly and quantitatively detect serum WFA+-IgA1. As previously reported, our CLEIA system realizes highly sensitive and rapid detection by immobilizing WFA at a high density on magnetic particles to form multimers. In addition, oligomerization of IgA1 in the blood accelerates the high avidity of WFA with its glycan epitopes. WFA is a lectin that mainly recognizes terminal GalNAc but also binds to multimerized sialic acid-deficient LacNAc [21]. The reactivity increases in various diseases, such as liver fibrosis, ovarian cancer, cholangiocarcinoma, and prostate cancer. For instance, WFA binds to the LacdiNAc structure (GalNAcβ1-3/4GlcNAc) containing the terminal GalNAc of *N*-glycans on prostate-specific antigens and ceruloplasmin in prostate cancer and ovarian cancer, respectively [38,40]. On the other hand, in liver fibrosis and cholangiocarcinoma, it has been suggested that WFA binds to multimerized sialic-acid-deficient LacNAc on M2BP and MUC1. In particular, M2BP is also a molecule that forms multimers. Thus, the development of automated CLEIA has been successful because of the multivalency of WFA and ligands [23]. Our results with automated detection clearly show that even for *N*-glycans, we can develop a diagnostic system that can be routinely detected by constructing an optimal sandwich detection system by considering the properties of lectins. Moreover, the construction of a suitable standard is important for the development of a reliable automated immunoassay system. Other groups have reported the construction of multiple standards for quantitative immunodetection [41]. In contrast, since we designed a suitable standard for automated CLEIA by constructing polymeric IgA1 for cross linking with glutaraldehyde, our standard construction is a simpler and more reliable approach compared to the report.

In a validation study to evaluate WFA+-IgA1 for IgAN diagnosis, although serum WFA+-IgA1 levels showed a significant difference between HCs and IgAN patients, no significant difference was observed between patients with and without IgAN (Figure 4). However, WFA+-IgA1 values in IgAN patients decreased significantly depending on the period after therapeutic intervention. Thus, although it remains possible that steroid interventions affect serum WFA+-IgA1 levels and further detailed investigation is needed, it is suggested that WFA+-IgA1 is involved in renal disease associated with IgA1. Abnormal IgA1, including agglutinated IgA1, is also present in patients with non-IgAN, such as asymptomatic IgA deposition. Based on the WFA multivalency described above, the increase in WFA+-IgA1 in some non-IgAN patients suggests that WFA recognizes abnormally agglutinated IgA1. Furthermore, abnormal glycosylation of IgA1 has been reported in healthy subjects [42]. There were a few non-IgAN patients in the present study and they included various diseases. Further investigations using larger cohorts are needed. As the other glycosylated IgA1, serum Gd-IgA1 levels were also measured to compare with WFA+-IgA1. The clinical performance of Gd-IgA1 for IgAN diagnosis was superior to that of WFA+-IgA1 in terms of AUC score. The diagnostic performance of Gd-IgA1 for IgAN was remarkably high compared with that of other serological indicators. However, it remained unsatisfactory in terms of sensitivity, because the Gd-IgA1 values of many patients with IgAN are relatively low. The IgA immune complex is formed by the production and binding of autoimmune antibodies against aberrant glycoforms in the hinge region of IgA1 [43,44]. Therefore, the low sensitivity may be caused by autoimmune antibodies that compete with the Gd-IgA1 antibody or occupy the glycopeptide antigen of the Gd-IgA1 antibody. Therefore, it is essential to develop a serological marker that surpasses or complements Gd-IgA1 levels to improve the serological diagnostic performance of IgAN. Although it is challenging to detect the disease relatively early with a single marker, if there are markers with different characteristics, it seems feasible to combine them to improve the diagnostic accuracy. Interestingly, the value of WFA+-IgA1 did not correlate with any serological markers, such as Gd-IgA1 or total IgA. This result clearly indicates that WFA+-IgA1 is an indicator independent of Gd-IgA1 and total IgA. Thus, as shown in Table 2, the combined diagnosis of WFA+-IgA1, Gd-IgA1, and C3 appears to improve the diagnostic accuracy of Gd-IgA1. Although the evaluation of WFA+-IgA1 for IgAN diagnosis requires further study and improvement, there is a possibility that WFA+-IgA1 will be a marker that complements the diagnostic performance of Gd-IgA1.

In the present study, the detailed glycan structure of serum IgA1 recognized by WFA was not clarified. Most *N*-glycans of IgA1 derived from HCs are bi-antenaries with terminal sialic acid modifications [45]. WFA does not bind to these glycans. Moreover, although it is a glycoform analysis of monomeric IgA1, no predominant difference in *N*-glycoforms in IgA1 derived from HCs and IgAN patients has been reported [10]. Curiously, even glycan analysis of IgA1 derived from IgAN patients did not confirm the glycan structure recognized by WFA. However, it has been reported that sialic acid of serum IgA1 and tonsil lymphocytes producing IgA1 are relatively reduced [46]. In addition, it has also been reported that an oligomeric IgA1-containing dimer has an increased content of immature glycoforms, such as asialo and high-mannose structure, compared to monomeric IgA1 [45]. Increased serum WFA+-M2BP levels in liver fibrosis are due to the increase in tri- and tetra-antenary asialo-glycans recognized by WFA on M2BP as liver fibrosis progresses [47]. Therefore, it is considered that the IgA1 recognition mechanism by WFA is due to an increase in asialo-glycans on IgA1 that form a multimer. In contrast, abnormal agglutination of IgA1 in blood is a form of autoagglutination, and immune complexes also form by interaction with other glycoproteins, such as the other globulin families, likely IgG and IgM. Recent developments in protein–protein interaction (PPI) analysis using mass spectrometry and data analysis have contributed to the elucidation of proteins and PPI networks whose functions are still unknown [48,49]. Analysis of immune complexes in IgAN has revealed the involvement of globulin, as well as various glycoproteins in the formation of IgA1 immune complexes [41]. Therefore, it is possible that WFA comprises not only the glycans on the agglutinated IgA1, but also the glycans on the glycoprotein, which form an immune complex with IgA1 in the serum. These remaining issues should be confirmed by detailed glycan structural analyses of IgA1 recognized with WFA using mass spectrometry and comprehensive analysis, considering the interaction between IgA1 and other glycoproteins.

## 4. Materials and Methods

### 4.1. Specimens

Sera from 50 patients without any history of IgAN or non-IgAN kidney diseases that were negative for hepatitis B and C viruses and human immunodeficiency virus (HCs) were purchased from TRINA BIOREACTIVES (Naenikon, Switzerland). Serum samples of non-IgAN were obtained from various patient groups, including anti-neutrophil cytoplasmic antibody-related nephropathy, membranous nephropathy, diabetic nephropathy, nephrosclerosis, minimal change nephrotic syndrome, focal segmental glomerular sclerosis, membranoproliferative glomerulonephritis, Alport syndrome, and acute nephritis (*N* = 43). Serum samples were obtained from 116 IgAN patients, including 47 before and 69 after tonsillectomy, who were diagnosed with IgAN by biopsy. Each sample was divided into 2.0 mL aliquots in sterile cryotubes and immediately frozen at −80 °C until later use. The study protocol conformed to the ethical guidelines of the 1975 Declaration of Helsinki. All patients provided written informed consent for the serum analysis. The study protocol was approved by the ethical committees of Sysmex and Narita Memorial Hospital (protocol code 2019-107, 2 March 2020). The experiments in this study were carried out in accordance with approved guidelines.

### 4.2. Immunoprecipitation of IgA1

IgA1 was purified from serum using biotinylated anti-IgA1 mAb-conjugated streptavidin-coated magnetic beads. To prepare the biotin antibody, mouse anti-human IgA1 mAb (Clone B3506B4; SouthernBiotech, Birmingham, AL, USA) was reduced with 2-mercaptoethylamide hydrochloride (Nacalai Tesque, Kyoto, Japan) and then desalted using prepacked disposable PD-10 columns (Cytiva, Tokyo, Japan). The antibody was coupled with *N*-biotinyl-*N*′-[2-(*N*-maleimido) ethyl] piperazine (biotin-PE-maleimide, Dojindo Laboratories, Kumamoto, Japan). After coupling, the biotin anti-IgA1 mAb was desalted using a PD-10 column (Cytiva). For immunoprecipitation, 2 μg of biotinylated anti-IgA1 mAb was immobilized to 10 μL of streptavidin-coated magnetic beads (Dynabeads MyOne Streptavidin T1; bead size, 1.0 mm; Thermo Fisher Scientific, Watham, MA, USA) by incubating at room temperature for 1 h. After washing the conjugated beads with phosphate-buffered saline containing 0.05% Tween 20 (PBST) three times, serum specimens diluted in PBST were incubated with antibody-conjugated magnetic beads for 1 h at room temperature. After washing the beads with PBST, elution was performed with 0.1 M glycine-HCl (pH 2.5) and then neutralized immediately with 1 M Tris-HCl (pH 9.0). The obtained elution fraction was used as a sample for subsequent experiments, including electrophoresis and lectin microarray analysis.

### 4.3. Immunoprecipitation of Aggregated IgA1 with WFA

WFA precipitation of serum IgA1 was performed using biotinylated WFA. Briefly, biotinylated WFA was prepared by coupling WFA (Vector Laboratories, Burlingame, CA, USA) with 6-(biotinylamino) hexanoic acid N-hydroxysuccinimide ester (biotin-AC_5_-OSu, Dojindo Laboratories). Biotinylated WFA (2.5 μg of biotinylated WFA) was pre-conjugated to 10 μL streptavidin-coated magnetic beads (Dynabeads, Thermo Fisher Scientific) by incubation at room temperature for 1 h. After washing with PBST, the diluted serum specimens were incubated with WFA-conjugated magnetic beads for 1 h at room temperature. After washing the beads, elution was performed with 0.1 M glycine-HCl (pH 2.5) and then neutralized immediately with 1 M Tris-HCl (pH 9.0). The eluted fraction was used as the sample for subsequent experiments.

### 4.4. SDS-PAGE and Western Blotting

The obtained samples were separated by 3–8% gradient SDS-PAGE using a NuPAGE Tris-acetate pre-cast gel (Thermo Fisher Scientific) under non-reducing conditions. The gel was visualized by silver staining using 2D-Silver Stain Reagent II (COSMO Bio, Tokyo, Japan). For Western blotting, transfer of proteins to a polyvinylidene difluoride membrane was performed using an iBlot 2 Dry Blotting System (Thermo Fisher Scientific). After blocking the transferred membrane with BlockAce (KAC, Kyoto, Japan), the membrane was incubated with biotinylated anti-IgA1 mAb (1 μg/mL in Tris-buffered saline containing 0.05% Tween 20 (TBST) and 10% BlockAce) at room temperature for 1 h. After washing, the membrane was incubated with horseradish-peroxidase-conjugated streptavidin diluted 1/10,000 in TBST; Jackson Immune Research, West Grove, PA, USA) at room temperature for 40 min. Blotting signals were visualized using ECL Prime Western Blotting Detection Reagent (GE Healthcare, Chicago, IL, USA).

### 4.5. Lectin Microarray

Differential glycan analysis of purified serum IgA1 was performed using antibody-overlay lectin microarray. Glycan profiling was performed as previously described [16,21]. Briefly, purified serum IgA1 (equivalent to 5 μL of serum) was applied onto a lectin microarray slide (LecChip^TM^; GlycoTechnica Co., Yokohama, Japan). The binding specificities of the 45 lectins immobilized on the array slide and the printing pattern of the array are described in Appendix A, respectively. After overnight incubation at 20 °C, human serum polyclonal IgG (20 µg) was added and incubated for 30 min at 20 °C. After the slide was washed three times with PBS containing 1% Triton X-100 (PBSTx), biotinylated anti-IgA1 mAb solution (200 ng in PBSTx) was applied to the array and incubated for 1 h at 20 °C. After the slide was washed three times with PBSTx, 400 ng of Cy3-labeled streptavidin (GE Healthcare UK, Little Chalfont, UK) in PBSTx was added to the array slide and incubated for 30 min at 20 °C. The slide was washed with PBSTx and scanned using a GlycoStation Reader 1200 evanescent-field fluorescent scanner (GlycoTechnica). All data were analyzed using an Array-Pro Analyzer, version 4.5 (Media Cybernetics, Bethesda, MD, USA). The net intensity of each spot was calculated by subtracting the background value from the total signal intensity of three spots. The mean lectin signals of triplicate spots were normalized to the mean value of the 45 lectins immobilized on the array. The data were analyzed using a normalization procedure as described previously [50].

### 4.6. Construction of IgA1 Multimer for Standard IgA1

Native human IgA (Abcam, Cambridge, UK) was cross linked by incubation for 1 h at 25 °C with glutaraldehyde (glutaraldehyde:IgA = 12,500:1 (mol/mol), pH 9.0). After incubation, the reactive solution was desalted using prepacked disposable PD-10 columns. For sialic acid digestion, neuraminidase (Nacalai Tesque) was added to the elution fraction and incubated for 1 h at 37 °C. To purify polymeric IgA, the reactive products were subjected to SEC with HiLoad 16/60 Superdex 200 prep grade (Cytiva) (Appendix A) or TSKgel G3000SWXL (TOSOH, Tokyo, Japan). The peak fractions corresponding to polymeric IgA were collected.

### 4.7. Measurement of Serum WFA+-IgA1 with the HISCL Automated Chemiluminescent Enzyme Immunoassay System

Serum WFA+-IgA1 measurements were performed with the HISCL automated CLEIA analyzer based on a previous report [23]. Briefly, serum WFA+-IgA1 was captured using WFA-conjugated streptavidin-coated magnetic beads. The captured product was detected using alkaline-phosphatase-conjugated anti-IgA1 monoclonal antibody. Using these reagents, serum WFA+-IgA1 levels were measured using an automated CLEIA analyzer, HISCL-5000 (Sysmex, Kobe, Japan). Serum specimens (10 µL) were diluted with 50 µL of reaction buffer (HISCL^TM^M2BPGi^TM^ R1 reagents, Sysmex). After incubation with a WFA-conjugated magnetic bead solution (HISCL^TM^M2BPGi^TM^ R2 reagents, Sysmex), 100 μL of alkaline phosphatase-labeled anti-human IgA1 monoclonal antibody diluted in dilution buffer (modified HISCL^TM^M2BPGi^TM^ R3 reagents, Sysmex) was added to the beads. After washing the beads, chemiluminescent reaction was performed using the HISCL Substrate Reagent Set (Sysmex). Chemiluminescent intensity was acquired within 17 min, as shown in Appendix A. The reaction chamber was maintained at 42 °C.

### 4.8. Measurement of Serum Gd-IgA1, Total IgA, and C3

The levels of serum Gd-IgA1, total IgA1, and C3 were measured using commercially available assay kits for Gd-IgA1 (Immuno-Biological Laboratories, Fujioka, Japan), IgA (N-Assay TIA Nittobo; Nittobo Medical, Tokyo, Japan), and C3 (Nittobo Medical), respectively. Each protocol was performed in accordance with the manufacturer’s instructions.

### 4.9. Statistical Analyses

Comparison of clinicopathological data between healthy donors, non-IgAN patients, and IgAN patients was evaluated using the Mann–Whitney *U* test. For combinational analysis of WFA+-IgA1 and other markers in Table 2, the following formulae were used:WFA-Gd combination = [WFA+-IgA1 (ng/mL)] + (30 × [Gd-IgA1 (ng/mL)])
WFA-Total combination = [WFA+-IgA1 (ng/mL)] + (4 × [Total IgA (mg/dL)]
WFA/C3 combination = [WFA+-IgA1]/(4 × [C3 (mg/dL)])
WFA-Gd/C3 combination = [WFA+-IgA1 (ng/mL)] + (30 × [Gd-IgA1 (ng/mL)])/(4 × [C3 (mg/dL)])

To assess the diagnostic performance in discriminating between samples from patients with IgAN and other diseases, including HCs, ROC curve analysis was performed to determine the AUC values. All calculations and graphical drawings were performed using GraphPad Prism version 6.00 for Windows (GraphPad Software, San Diego, CA, USA).

## 5. Conclusions

We focused on the *N*-glycosylation of serum IgA1 and constructed a robust automated sandwich immunoassay system using lectins. Our results indicate that the *N*-glycosylation of glycoproteins, which is difficult to detect with a general immunoassay system, can be detected quantitatively and with high sensitivity. The main purpose of this study was to adapt IgA1 modified with aberrant *N*-glycosylation to an automatic immunoassay system and biological analysis of the WFA recognition mechanism. In this study, we succeeded in constructing a robust immunoassay system using lectins, but unfortunately, WFA+-IgA1 did not show useful clinical performance compared to other markers. The ultimate goal is to develop diagnostic markers that can be used clinically for routine use. To achieve this goal, it is important to perform detailed glycan structural and omics analyses, including PPIs, together with the development of an automated immunoassay system. The ultimate advantage of our CLEIA system is that rapid and highly sensitive detection can be realized quantitatively and reproducibly. Although some considerations remain, we believe that the achievements of this study highlight the feasibility for the clinically practical application of glycoprotein diagnostic markers that focus on aberrant glycosylation.

## Figures and Tables

**Figure 1 ijms-23-05165-f001:**
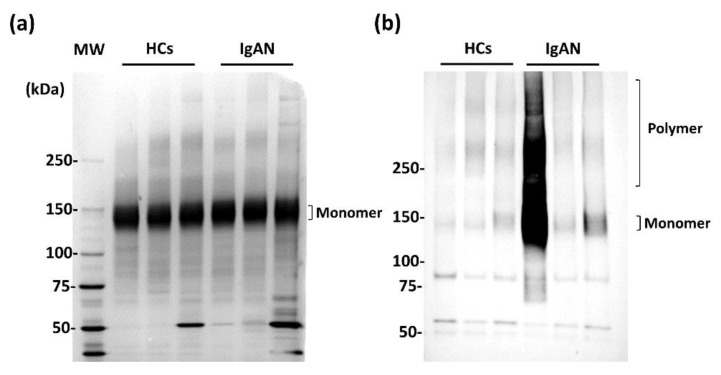
Verification of WFA binding to serum-agglutinated IgA1. (**a**) Silver stain of immunoprecipitated fraction with anti-IgA1 mAb. (**b**) Western blot analysis of WFA-immunoprecipitated fraction detected with anti-IgA1 mAb. MW; molecular weight, HCs; healthy control subjects.

**Figure 2 ijms-23-05165-f002:**
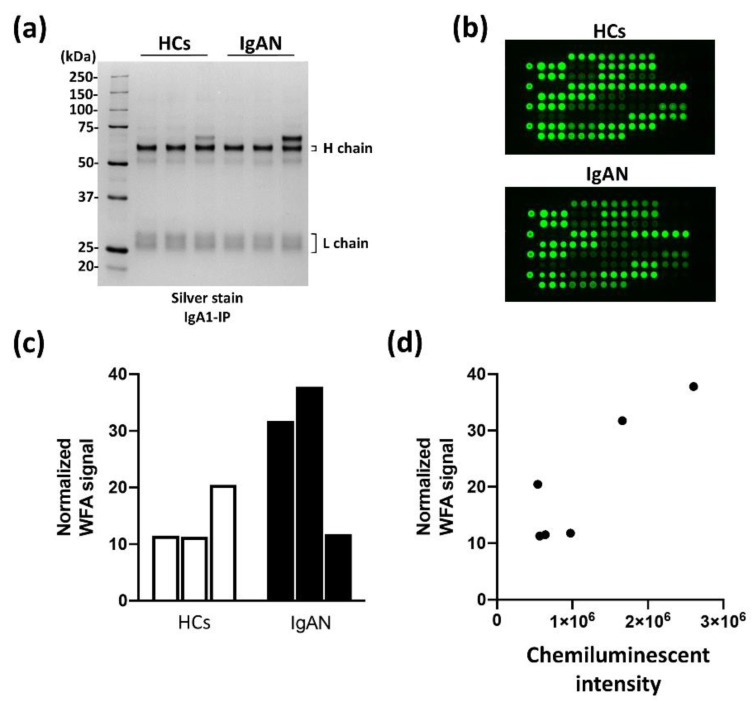
Correlation between glycan profiling of IgA1 and WFA+-IgA1 CLEIA. (**a**) SDS-PAGE and silver staining of IgA1 purified by immunoprecipitation from serum. Electrophoresis was performed under the reducing condition. (**b**) Typical images of signals of 45 lectins in the lectin microarray analysis of HC and IgAN patients. (**c**) WFA signals of each IgA1 derived from three HC and three IgAN patients. (**d**) Two-dimensional plot of WFA signals and chemiluminescent counts of WFA+-IgA1 CLEIA.

**Figure 3 ijms-23-05165-f003:**
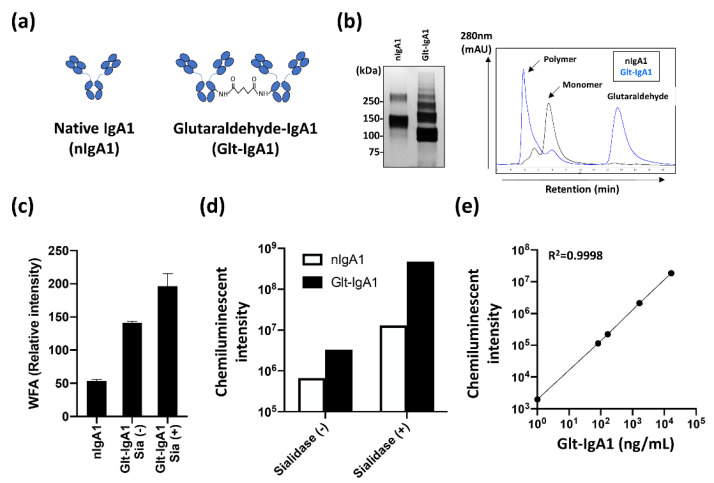
Construction and evaluation of standard IgA1 for automated immunoassay system. (**a**) Illustrations of mono (nIgA1) and polymeric IgA1 (Glt-IgA1) standards. (**b**) Evaluation of each IgA1 standard. SDS-PAGE and silver stain were performed under the non-reducing condition. The standards were analyzed by size exclusion chromatography. Black and blue lines indicate nIgA1 and Glt-IgA1, respectively. (**c**) WFA signals evident in differential glycan analysis with antibody-overlay lectin microarray of standards. nIgA1 represents without sialidase digestion Glt-IgA1 Sia (+) and Sia (−) denotes presence and absence, respectively, of sialidase digestion. (**d**) Differential measurements of constructed standard IgA1 by HISCL an automated chemiluminescent immunoassay system. Each standard was compared with (+) or without (−) sialidase digestion. (**e**) Linear responsibility of chemiluminescent count and concentration of Glt-IgA1 with sialidase digestion.

**Figure 4 ijms-23-05165-f004:**
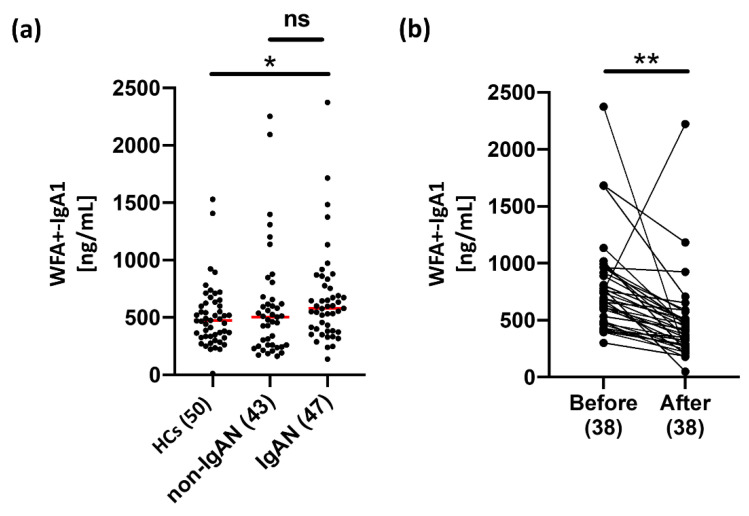
Measurement of serum WFA+-IgA1 levels with automated CLEIA. (**a**) Serum WFA+-IgA1 levels in HCs (*N* = 50), non-IgAN patients (*N* = 43), and IgAN patients (*N* = 47). (**b**) Comparison of serum WFA+-IgA1 levels before and after tonsillectomy in IgAN patients (*N* = 38). Values that significantly differed between each subject group were identified by the Mann–Whitney U test. The medians of each group are shown as the red bar. * *p* < 0.05, ** *p* < 0.01, ns; not significant.

**Table 1 ijms-23-05165-t001:** Clinical features of each marker in HS, non-IgAN, and IgAN patients.

	HCs (*N* = 50)	Non-IgAN (*N* = 43)	IgAN (*N* = 47) ^1^	HCs vs. IgAN	non-IgAN vs. IgAN
IgA (mg/dL)	212.0 ± 109.9	218.0 ± 121.4	270.0 ± 108.8	*p* < 0.01	*p* < 0.05
C3 (mg/dL)	156.0 ± 409.6	116.0 ± 23.8	113.5 ± 21.0	*p* < 0.01	ns ^2^
IgA/C3	1.32 ± 23.8	1.95 ± 23.8	2.28 ± 23.8	*p* < 0.01	*p* < 0.05
Gd-IgA (ng/mL)	23.2 ± 14.0	22.2 ± 14.3	33.5 ± 14.7	*p* < 0.01	*p* < 0.01
WFA+-IgA1 (ng/mL)	474.5 ± 271.1	506.0 ± 468.0	579.4 ± 468.0	*p* < 0.05	ns

^1^ 47 cases of IgAN patients before tonsillectomy. ^2^ ns, not significant. Each value represents median ± standard deviation.

**Table 2 ijms-23-05165-t002:** Comparative analysis of single and combined markers for IgAN diagnosis.

Marker(s)		AUC (95% CI)	*p* Value	Sensitivity (%) *	Specificity (%) *
Single marker	WFA+-IgA1	0.634 (0.523–0.745)	0.0234	66.0	62.0
	Gd-IgA1	0.734 (0.649–0.819)	<0.0001	72.3	69.6
	Total IgA	0.670 (0.579–0.762)	0.0011	89.4	43.5
	C3	0.709 (0.624–0.793)	<0.0001	87.5	52.7
Combination	WFA-Gd	0.748 (0.651–0.820)	<0.0001	74.5	67.7
	WFA-Total	0.660 (0.569–0.752)	0.0021	80.9	51.7
	WFA/C3	0.704 (0.616–0.792)	<0.0001	78.7	62.6
	WFA-Gd/C3	0.789 (0.714–0.865)	<0.0001	83.0	70.3

* The cutoff values obtained from Youden’s index. AUC, area under the ROC curve; WFA-Gd, combination of WFA+-IgA1 and Gd-IgA1; WFA-Total, combination of WFA+-IgA1 and total IgA; WFA/C3, combination of WFA+-IgA1 and C3; WFA-Gd/C3, combination of WFA+-IgA1, Gd-IgA1, and C3; CI, confidence interval.

## Data Availability

Data available on request from corresponding author.

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
