# Peer review of "Development and Evaluation of a Robust Sandwich Immunoassay System Detecting Serum WFA-Reactive IgA1 for Diagnosis of IgA Nephropathy"

_ijms, 2022, doi:10.3390/ijms23095165_

Round 1

Reviewer 1 Report

In this manuscript, the authors developed a new test WFA reactive IgA1 which showed increased WFA-IgA1 complex in IgA nephropathy patients compared to healthy individuals.

Does this test work on other fluids like urines?

Did the authors test different doses of WFA? Is it a saturating concentration?

For ROC curve, the authors only described AUC. They should add CI and p. A cut off value would be also informative with negative and positive predictive values.  

Author Response

We are grateful to reviewer fot the critical comments and useful suggestions that have helped us to improve our paper considerably.  As indicated in the responses that follows, we have taken all these commnets and suggestions into account in the revised version of our paper.

Point 1: Does this test work on other fluids like urines?

Response 1: The usefulness of this marker in urine has not yet been investigated. The assessment of this marker in urine will be carried out as the next plane. So, we added a sentence to the conclusion “a further investigation using urine specimens will be carried out”.

Point 2: Did the authors test different doses of WFA? Is it a saturating concentration?

Response 2: We performed the sandwich immunoassay under the saturated condition of WFA to detect the abnormal agglutinated IgA1 in serum.

 Point 3: For ROC curve, the authors only described AUC. They should add CI and p. A cut off value would be also informative with negative and positive predictive values.

 Response 3: As your suggestion, Table 2 in page 7 has been modified with 95% confidence interval and P values.

Reviewer 2 Report

The authors showed development and validation of an immunoassay for N-glycosylation of serum IgA1  using lectins. However, they failed to demonstrate its clinical usefulness for the routine use in daily clinical practice. I disagree that a larger cohort will change anything.

In my opinion the conclusions should be limited to the specificity of the new assay. In addition it should be stated that this assay will not improve the diagnosis of IgA nephropathy.

Methodology and results are clearly described.

Major comments – the conclusion has to be changed.

Minor:

  1. The sentence in the introduction ‘However, even with this antibody, it is still difficult to accurately and sensitively diagnose IgAN.’ should be supplemented ‘without kidney biopsy’
  2. Please consider changing ‘normal’ control subjects to ‘healthy control subjects’ throughout the manuscript.

Author Response

We are grateful to reviewer fot the critical comments and useful suggestions that have helped us to improve our paper considerably.  As indicated in the responses that follows, we have taken all these commnets and suggestions into account in the revised version of our paper.

Point 1: The authors showed development and validation of an immunoassay for N-glycosylation of serum IgA1 using lectins. However, they failed to demonstrate its clinical usefulness for the routine use in daily clinical practice. I disagree that a larger cohort will change anything. In my opinion the conclusions should be limited to the specificity of the new assay. In addition, it should be stated that this assay will not improve the diagnosis of IgA nephropathy.

Response 1: As suggested in the Reviewer’s comment, in this study, we succeeded in constructing a new assay using lectin, but we have not shown its clinical significance in the diagnosis of IgAN. Therefore, in the conclusion, add a reference to the fact that there was no clinical performance, and change to mention only that a new assay system could be constructed. In addition, the sentence “further verification using a much larger cohort” was deleted.

Point 2: The sentence in the introduction ‘However, even with this antibody, it is still difficult to accurately and sensitively diagnose IgAN.’ should be supplemented ‘without kidney biopsy’

Response 2: As your suggestion, we added the sentence “without kidney biopsy” to the end of the sentence on line 75 of the introduction.

Point 3: Please consider changing ‘normal’ control subjects to ‘healthy control subjects’ throughout the manuscript.

Response 3: As your suggestion, all the words “normal control subjects” in the manuscripts including the Figure was changed to “healthy control subjects”.

Round 2

Reviewer 2 Report

The response and correction are appropriate. The conclusion was modified and reflects the data. No further comments.